# Cardiovascular outcome in 12-month-old male and female offspring of metformin-treated obese mice

Josca M. Schoonejans[1,6] , Phoebe Wilsmore[1] , Lais V. Mennitti[1] , Kwun K. Wong[1] ,
Thomas J. Ashmore[1], Tessa A. C. Garrud[2] , Heather L. Blackmore[1] , Olga V. Patey[2] ,
Denise S. Fernandez-Twinn[1] , Dino A. Giussani[2,3,4,5] and Susan E. Ozanne[1,3,4,5]

[1]*Institute of Metabolic Science-Metabolic Research Laboratories and MRC Metabolic Diseases Unit, University of Cambridge, Cambridge, UK*
[2]*Department of Physiology, Development and Neuroscience, University of Cambridge, Cambridge, UK*
[3]*Cambridge BHF Centre of Research Excellence, Cambridge, UK*
[4]*Cambridge Reproduction Interdisciplinary Centre, Cambridge, UK*
[5]*The Loke Centre for Trophoblast Research, Cambridge, UK*
[6]*Institute of Reproductive Biology, Department of Metabolism, Digestion and Reproduction, Imperial College, London, UK*

Handling Editors: Laura Bennet & Christopher Lear

The peer review history is available in the Supporting Information section of this article (https://doi.org/10.1113/JP288696#support-information-section).

J. M. Schoonejans is eligible for the Early Investigator Prize.

The Journal of Physiology

**Abstract figure legend** This study described the effects of exposure to maternal obesogenic diet and metformin treatment during pregnancy on offspring cardiovascular outcomes. Maternal diet-induced obesity and glucose intolerance led to predominantly vascular and cardiac abnormalities in 12-month-old female and male offspring, respectively. This phenotype was not corrected by maternal metformin treatment during gestation. Indeed, the intervention programmed independent cardiovascular alterations in offspring of both sexes. This highlights the need for sex-dependent follow-up and warrants further study into longer-term cardiovascular effects in human children from mothers treated with metformin during pregnancy. Created with BioRender.com. DT, deceleration time; E/A = ratio between early (E) diastolic and atrial contraction-related diastolic filling (A) velocities; IVRT, isovolumetric relaxation time; MPI, myocardial performance index.

**Abstract** Metformin is increasingly used to treat diabetes in pregnancy, but the effects on adult offspring health remain under-explored. The present study investigated the long-term cardiovascular effects in male and female offspring of maternal metformin treatment using a well-established mouse model of obese glucose intolerant pregnancy. Female mice were given chow, or an obesogenic diet with/without 300 mg kg$^{-1}$ day$^{-1}$ oral metformin during gestation. At 3, 6 and 12 months of age, male and female offspring were studied longitudinally with tail-cuff plethysmography and echocardiography. At 12 months, tissues were collected for wire myography, histology and molecular analyses. Female offspring of obese dams had elevated blood pressure throughout life, cardiac diastolic dysfunction at 3 months, and increased femoral vasoconstrictor reactivity and aortic wall remodelling at 12 months. Metformin treatment did not ameliorate these effects and led to obesity-induced hypertension at 12 months. Irrespective of metformin, male offspring of obese pregnancy had cardiac diastolic dysfunction from 6 months without changes in blood pressure. Male metformin-exposed offspring also showed cardiomegaly, increased cardiac collagen and vascular sympathetic hyperreactivity, suggesting metformin exposure worsened the cardiovascular phenotype. These findings show that maternal obesity caused sex-specific cardiovascular aberrations in aged offspring. Maternal metformin was not corrective and introduced further sex-dependent cardiovascular alterations. Further long-term offspring follow up of both sexes is needed for informed decisions about metformin during pregnancy.

(Received 6 February 2025; accepted after revision 24 May 2025; first published online 19 June 2025)

**Corresponding authors** J. M. Schoonejans: Institute of Reproductive Biology, Department of Metabolism, Digestion and Reproduction. Hammersmith Hospital, Imperial College London, London W12 0NN, UK. Email: josca.schoonejans11@imperial.ac.uk, S. E. Ozanne: Institute of Metabolic Science-Metabolic Research Laboratories and MRC Metabolic Diseases Unit, Addenbrooke's Hospital, University of Cambridge, Cambridge, CB2 OQQ, UK. Email: seo10@cam.ac.uk

## Key points

- The oral medication metformin is increasingly used to treat diabetes in pregnancy.
- Metformin readily crosses the placenta, and long-term effects on offspring cardiovascular health remain unexplored in human and animal studies.
- In a mouse model of maternal diet-induced obesity with impaired glucose tolerance, female and male offspring developed hypertension and diastolic cardiac dysfunction, respectively, by 12 months of age (equivalent to middle age in humans).
- Maternal metformin treatment worsened the cardiovascular phenotype and introduced further sex-dependent cardiovascular alterations in both male (cardiac stiffening, vascular dysfunction) and female (obesity-induced hypertension) offspring.
- This work highlights that long-term cardiovascular follow up in offspring of both sexes from human pregnancies treated with metformin is crucial to make more informed decisions about metformin use in diabetic pregnancy.

## Introduction

Almost half of women worldwide currently enter pregnancy overweight or obese (Kent et al., 2024). Obesity in pregnancy is associated with adverse pregnancy outcomes including the development of gestational diabetes mellitus (GDM), hypertension in pregnancy, and even stillbirth (Davies, 2015). The incidence of GDM has been increasing and it is estimated to affect one in seven pregnancies worldwide (Saeedi et al., 2021). Metformin, an oral biguanide, is the most prescribed glucose-lowering drug in diabetic pregnancies globally (Cesta et al., 2019), and its use during pregnancy is being investigated for other indications including polycystic ovary syndrome (PCOS) (Hanem et al., 2019), glucose tolerant obesity (Syngelaki et al., 2016) and pre-eclampsia (Cluver et al., 2021). The use of metformin in pregnancies complicated by GDM is deemed safe, improves maternal glycaemic control, and is associated with benefits including decreased gestational weight gain, lower incidence of macrosomia, pre-eclampsia and neonatal intensive care unit admission (Guo et al., 2019). However, around one-third of women do not achieve glucose control with metformin alone and many require supplemental insulin treatment (Tarry-Adkins et al., 2020). Importantly, metformin rapidly crosses the human placenta and is found at similar concentrations in maternal and fetal blood (Priya & Kalra, 2018). Therefore, metformin could exert direct effects on the developing fetus.

Maternal obesity and diabetes during pregnancy are associated with adverse long-term cardiometabolic outcomes in offspring in both humans and animal models (Schoonejans & Ozanne, 2021), sparking interest in potential intervention strategies to prevent transmission of poor cardiometabolic health between mother and child (Cochrane et al., 2024). There are limited studies in either humans or preclinical animal models that address potential beneficial or detrimental consequences of gestational metformin treatment on long-term offspring health. There is evidence of increased adiposity in children of metformin-treated women with GDM or PCOS compared to insulin or placebo, respectively (Hanem et al., 2019; Tarry-Adkins et al., 2019), but, to our knowledge, only two studies have investigated cardiovascular outcomes in humans. Metformin administration in glucose tolerant obese pregnancies was associated with subtly altered haemodynamic and diastolic parameters in 4-year-old children (Panagiotopoulou et al., 2020), changes that, if persistent, may be beneficial for future cardiovascular health. In contrast, adverse outcomes were seen in 5–10-year-old children of metformin-treated mothers with PCOS, such as increased risk of 'metabolically unhealthy' obesity (a composite outcome that included hypertension) and increased heart rate in male children (Hanem et al., 2019). However, children from these studies remain young, and further research is required to determine ageing-dependent effects in adult individuals prenatally exposed to metformin. Animal models are therefore crucial in determining the longitudinal long-term effects of maternal metformin treatment on offspring cardiovascular outcomes, with the aim to inform human studies. Using a well-established mouse model of diet-induced maternal obesity and glucose intolerance, we have studied previously the effect of maternal metformin treatment: metformin treatment ameliorated maternal glycaemia, adiposity and hepatic steatosis, but did not rescue the phenotype of fetal growth restriction and catch-up growth in offspring (Hufnagel et al., 2022; Schoonejans et al., 2021). In 12-month-old offspring, *in utero* metformin exposure led to increased adiposity, adipose tissue inflammation, hepatic steatosis and insulin resistance (Schoonejans et al., 2022), indicating adverse effects on cardiovascular risk factors. No study has investigated cardiovascular function in aged adult offspring of metformin-treated compromised pregnancies.

Therefore, the present study aimed to investigate the cardiovascular effects of maternal metformin intervention during pregnancy on male and female offspring until 12 months of age (equivalent to middle age in humans), using a well-established mouse model of diet-induced obese, glucose intolerant pregnancy. We adopted an integrative approach combining *in vivo* longitudinal investigation of arterial blood pressure and cardiac function with experiments in isolated vessels and tissues using cellular and molecular approaches to address potential underlying mechanisms at the organ level.

**Josca Schoonejans** holds a PhD from the University of Cambridge, where she studied the effect of maternal metformin treatment on young and aged offspring, focusing on age- and sex-specific negative effects on adipose tissue function and cardiometabolic health. She is currently a research fellow at Imperial College London and has secured funding to carry out her independent research into novel treatments for maternal metabolic diseases during pregnancy and their effects on fetal growth and adipose tissue development.

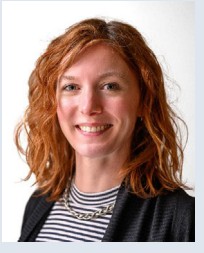

## Methods

### Ethical approval

All mouse work was performed according to the Home Office Animals (Scientific Procedures) Act 1986 Amendment Regulations 2012, after ethical review by the University of Cambridge Animal and Welfare Ethical Review Board (PP8498895). The authors understand the ethical principles under which *The Journal of Physiology* operates, and this work complies with the ARRIVE Guidelines for reporting *in vivo* experiments.

### Animal model

Female C57Bl/6J mice (Charles River Laboratories, Harlow, UK; RRID:IMSR_JAX:000664, bred inhouse) were randomly assigned, in numbers based on need to equalise numbers, by a technician not involved in the research, to receive either a standard chow diet [control (Con)] or a high-fat, high-sugar obesogenic diet [obese (Ob) and obese metformin-treated (Ob-Met) dams], *ad libitum*, from weaning up to and including pregnancy and lactation. The control diet was RM1 (7% sugars, 3% fat, w/w; Special Diets Services, Witham, UK). The obesogenic diet was composed of high-fat pellets (10% sugars, 20% fat, w/w; Special Diets Services, Witham, UK) and sweetened condensed milk [55% sugar, 8% fat, w/w; Nestle, Crawley, UK) supplemented with a vitamin and mineral pre-mix (AIN-93G-MX; Special Diets Services, Witham, UK). Dams were mated for a first pregnancy when on the experimental diet, and only proven breeders were used for the second (experimental) pregnancy when an adiposity threshold was reached (>12 g or <5 g for females fed the obesogenic diet or control diet, respectively). Detailed body weight data for the dams in this study were published previously (Schoonejans et al., 2021). In this model, Ob dams are glucose intolerant during their second pregnancy (Fernandez-Twinn et al., 2017), but not after (Furse et al., 2021). Ob-Met dams received 300 mg kg$^{-1}$ day$^{-1}$ metformin orally from one week prior to mating until day 19 of this experimental second pregnancy. This dose is clinically relevant, equating to 1700 mg in a 70 kg human adult (Salomäki et al., 2013) and leads to maternal serum metformin concentrations comparable to those observed in pregnant women (Hufnagel et al., 2022). No metformin only group was included as metformin is not given in healthy pregnancy; therefore, inclusion of this group is not clinically relevant. All offspring were weaned onto standard RM1 chow fed *ad libitum* and housed in same-sex littermate pairs. At 12 months of age, offspring were killed by exposure to rising $CO_2$ concentrations after a 16 h overnight fast (sibling used for wire myography and molecular analyses, transferred to clean cage) or cervical dislocation in the fed state (sibling used for echocardiography, plethysmography and aortic histology, kept in home cage). Random assignment of siblings to each group was carried out by an animal technician not involved in cardiovascular phenotyping. Animals in all three groups were bred concurrently (not sequentially). The body composition and metabolic phenotype of these animals have been published previously (Schoonejans et al., 2022). The animal model is outlined in Fig. 1.

Only one offspring (of each sex) from a particular dam was included in each experiment, therefore the experimental unit in this study was the dam. Each dam provided only one litter. Offspring numbers were 12 per group for the sibling undergoing physiology experiments based on previous work (Beeson et al., 2018) [tail cuff plethysmography and echocardiography, actual numbers vary between six and 12 as a result of quality control or statistical outliers (Grubb's method), numbers for each are indicated in relevant figure legends] and 11 or 12 for the sibling generating tissues for wire myography and molecular work, except male Ob offspring for which this was eight. One Ob-Met male offspring was culled at three months of age prior to the end of the study because of an adverse reaction to the anaesthesia; therefore, in total, 13 male offspring were bred for physiological experiments in this group. Where possible male and female offspring were taken from a single dam to reduce the number of dams required. The total number of dams used in this study was 54 (18 control, 21 obese and 15 obese metformin). The total number of offspring was 130.

### Non-invasive tail cuff plethysmography

Serial arterial systolic blood pressure (SBP) measurements were made in the same conscious animal using non-invasive restraint tail cuff plethysmography (BP-200 system; Visitech, Drammen, Norway) at 3, 6 and 12 months of age. Animals were subject to two days training and a minimum of 10 SBP and heart rate (HR) measurements were recorded on the third day. All sessions were performed by the same researcher at the same time of day (16.00 h to 17.00 h) to control for the natural circadian rhythm of arterial blood pressure. Data were only used if more than 10 measurements could be obtained and within-animal variation for that timepoint was less than 15%.

### Echocardiography

*In vivo* left ventricular (LV) function was assessed longitudinally by transthoracic echocardiography (Vevo 3100 system with MX400 probe; Fujifilm Visualsonics, Toronto, ON, Canada) under short-term general anaesthesia by isoflurane inhalation (induction at 2%

isoflurane, maintenance between 1.5% and 2%) at 3, 6 and 12 months of age. Mice were placed in the supine position on a heated platform. Body temperature was monitored using a rectal probe. Echocardiography images were obtained in the parasternal long-axis view (PSLAX) and apical four-chamber view, as recommended (Lindsey et al., 2018). LV geometry was assessed using B-mode imaging of the whole left heart and M-mode imaging of the mid-cardiac section (for measurements, see Table 1). PSLAX B-mode LV tracing was also used to determine LV systolic function (for calculations, see Table 1). Mitral valve flow was recorded using pulsed wave Doppler in the apical four-chamber view to determine indices of LV diastolic function (Table 1). Data from recorded clips were analysed using the Cardiac Package from VevoLAB, version 3.2.6 (Fujifilm Visualsonics). Data were only analysed if HR was >400 beats min$^{-1}$ and body temperature $37.0 \pm 1.5°C$ (Lindsey et al., 2018). A subset of data was analysed by two independent observers to control for observer bias and no difference was observed between them.

### Wire myography

Second-order femoral arteries were dissected and placed in Krebs buffer (NaCl, 118.5 mM, S7653, Sigma, St Louis, MO, USA; NaHCO$_3$, 25.0 mM, S5761, Sigma; KCl, 4.7 mM,

P5405, Sigma; MgSO$_4$·7H$_2$O, 1.2 mM, 101514Y, BDH, Poole, UK; KH$_2$PO$_4$, 1.2 mM, P9791, Sigma; CaCl$_2$, 2.5 mM, C7902, Sigma; glucose, 2.8 mM, G8270, Sigma). A 2 mm arterial segment was mounted in a two-chamber small-vessel wire myograph (Dual-Channel Wire Myography System 420A; DMT, Hinnerup, Denmark) using 40 μm diameter stainless steel wire, taking care to not damage the endothelium, as standard. The chamber was continually gassed with 5% CO$_2$ and 95% O$_2$ and maintained at 37°C. One myograph jaw was connected to a micrometre screw and the other to a force transducer connected to a data acquisition system (LabChart 8; ADInstruments, Dunedin, New Zealand). After a 20 min stabilisation period, resistance vessels were subjected to an established normalisation protocol consisting of alternating exposure to a modified Krebs buffer containing 64 mM K$^+$ (in exchange for Na$^+$ to maintain osmolality) and the jaws being moved apart until the magnitude of contraction elicited by exposure to 64 mM K$^+$ plateaued. This achieved an optimal internal vessel diameter for myography (typically 200–250 μm) and working tension equivalent to a physiological transmural pressure (Herrera et al., 2010; Skeffington et al., 2016). Vessels were then exposed to successive Krebs solutions with increasing K$^+$ concentrations (16–125 mM) to generate a K$^+$ concentration-response curve. Cumulative dose-response curves to phenylephrine (PE), sodium

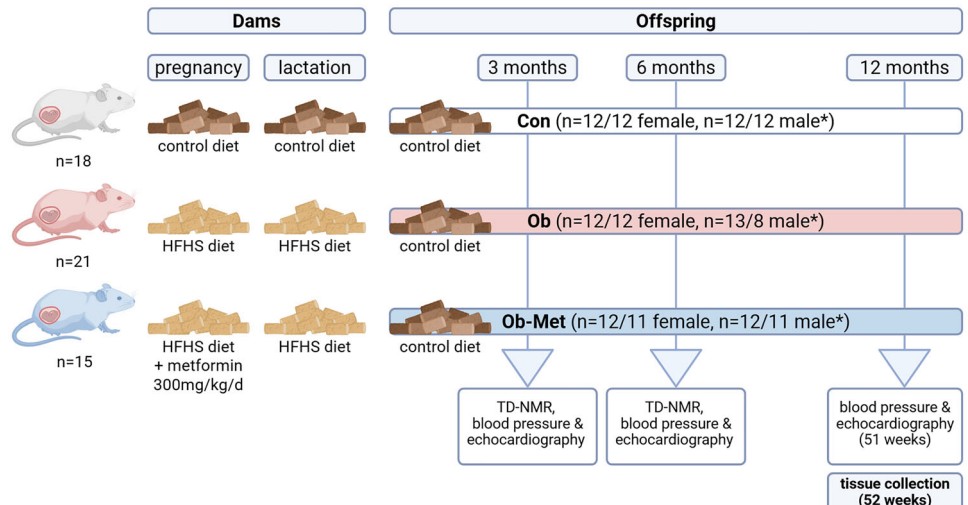

**Figure 1. Animal model**
Con = offspring from control-fed dams; Ob, offspring from dams fed obesogenic diet; Ob-Met, offspring from dams fed obesogenic diet treated with 300 mg kg$^{-1}$ day$^{-1}$ metformin according to previously published work (Schoonejans et al., 2021, 2022). HFHS, high-fat high-sugar diet. TD-NMR, time-domain nuclear magnetic resonance. *Offspring *n* = *a/b* refers to sibling pairs for which *n* = *a* siblings were used for *in vivo* physiological measurements and histology and *n* = *b* siblings for wire myography and molecular analyses. Created with BioRender.com.

**Table 1. Echocardiography measurements and calculations**

| View/tool | Measurements | Calculation |
|---|---|---|
| PSLAX B-mode<br>LV trace | • EDA<br>• ESA<br>• EDV*<br>• ESV*<br>• HR | • $SV = EDV - ESV$<br>• $CO = \frac{SV \times HR}{1000}$<br>• $EF = 100 \times \frac{EDV - ESV}{EDV}$ |
| PSLAX B-mode<br>Linear mode | • EDL<br>• ESL | • FS (longitudinal) $= 100\% \times \frac{EDL - ESL}{EDL}$<br>• Sphericity index $= \frac{EDV}{(\frac{4}{3} \times \pi r^3) \times 100\%}$ |
| PSLAX M-mode<br>Linear mode | • EDD<br>• ESD | • FS (radial) $= 100\% \times \frac{EDD - ESD}{EDD}$ |
| Apical four-chamber<br>Pulse-wave Doppler | • AET<br>• NFT<br>• $IVCT_{ms}$<br>• $IVRT_{ms}$<br>• A wave peak velocity<br>• E wave peak velocity<br>• Deceleration slope<br>• DT | • $E/A = \frac{E\ wave}{A\ wave}$<br>• $MPI = \frac{NFT - AET}{AET}$ |
| Apical four-chamber<br>Manual | • Cycle length | • $IVCT_{\%\ cycle} = 100\% \times \frac{IVCT_{ms}}{cycle\ length}$<br>• $IVRT_{\%\ cycle} = 100\% \times \frac{IVRT_{ms}}{cycle\ length}$ |

*Formulas for B-mode LV trace volume estimation are based on rotational volume estimation methods not published by VevoLab. AET, aortic ejection time; CO, cardiac output; DT, deceleration time; EDA, end-diastolic area; EDL, end-diastolic length; EDD, end-diastolic diameter; EDV, end-diastolic volume; EF, ejection fraction; ESA, end-systolic area; ESL, end-systolic length; ESD, end-systolic diameter; ESV, end-systolic volume; FS, fractional shortening; HR, heart rate; $IVCT_{\%cycle}$, isovolumetric contraction time expressed as percentage of the cardiac cycle; $IVCT_{ms}$, isovolumetric contraction time in milliseconds; $IVRT_{\%cycle}$, isovolumetric relaxation time expressed as percentage of the cardiac cycle; $IVRT_{ms}$, isovolumetric relaxation time in milliseconds; NFT, non-filling time; PSLAX, parasternal long axis; SV, stroke volume.

nitroprusside (SNP) and ACh were then created in log increments. Vessels were washed with Krebs buffer and allowed to rest for ≥15 min between curves. For vasodilator curves (SNP and ACh), vessels were pre-constricted with 10–2 M noradrenaline. Myography data were randomised and blinded prior to analysis. Data were extracted from blinded LabChart files manually: for each curve, raw tension for each $K^+$ concentration and agonist dose was recorded at the point where the response plateaued. For $K^+$ and PE, tension data were expressed relative to the baseline vessel tension for that curve using the average resting tension before and after the dose–response curve (ΔmN). Response magnitude for each PE dose was expressed as the relative change in tension compared to the maximum response during the $K^+$ concentration-response curve of the same vessel (%$K^+$), as standard (Herrera et al., 2010; Skeffington et al., 2016). For SNP and ACh, tension data were expressed relative to both the baseline vessel tension for that curve

(100% relaxation) and vessel tension after pre-constriction with noradrenaline (0% relaxation).

### Gene expression

Ventricular tissue was excised from 12-month-old, 16 h fasted animals and snap frozen on dry ice. Ventricular tissue was pulverised by pestle and mortar in presence of liquid nitrogen and subsequently lysed on ice in QIAzol Lysis Reagent (Qiagen, Germantown MD, USA) using the TissueRuptor II (Qiagen). Total RNA was extracted using the miRNeasy MiniKit (Qiagen) according to manufacturer instructions. Assessment of RNA quality, quantity and cDNA synthesis was performed as described previously (Schoonejans et al., 2021). Real-time quantitative PCR was performed using the SYBR Green PCR Master Mix (Thermo Fisher Scientific, Waltham, MA, USA) and the QuantStudio 5 Real-Time PCR System (Thermo Fisher Scientific).

**Table 2. Primer sequences for real time quantitative PCR**

| Gene | Forward 5′- to 3′ | Reverse 5′- to 3′ |
| --- | --- | --- |
| *Col1a1* | ACGCCATCAAGGTCTACTGC | ACTCGAACGGGAATCCATCG |
| *Col3a1* | TGACTGTCCCACGTAAGCAC | GAGGGCCATAGCTGAACTGA |
| *Col4a1* | GGCCCTTCATTAGCAGGTGT | GTGAGGACCAACCGTTAGGG |
| *Gapdh* | GACGGCCGCATCTTCTTGT | CACACCGACCTTCACCATTTT |
| *Ppia* | GTCCAGGAATGGCAAGACCA | GGGTAAAATGCCCGCAAGTC |
| *Sdha* | TTACAAAGTGCGGGTCGATGA | TGTTCCCCAAACGGCTTCTT |

Data were analysed using the comparative CT method normalised against expression of the geometric mean of the housekeeper genes *Gapdh*, *Ppia* and *Sdha*, for which the expression did not change between groups. Data are expressed normalised against the control group. Primer sequences are shown in Table 2.

### Histological analysis

A segment of the aorta was fixed in 4% formalin. Embedded segments were transversely sectioned and stained with Picrosirius Red to enable clear distinction between the tunica media and adventitia. Whole sections were scanned using the Axioscan digital slide scanner (Zeiss, Oberkochen, Germany). Images were analysed using QuPath (Bankhead et al., 2017) and the average thickness of the tunica media was calculated using:

$$\text{average thickness} = \frac{2 \times \text{tunica media cross} - \text{sectional area}}{\text{outer perimeter} + \text{inner perimeter}}$$

Data were analysed by two independent observers to control for observer bias and no difference was observed between them.

### Serum analysis

Serum concentrations of hormones used in correlation analyses in the present study have been published previously (Schoonejans et al., 2022).

### Statistical analysis

All data were analysed using Prism, version 8 and 9 (GraphPad Software Inc., San Diego, CA, USA). Data are presented as the mean ± SEM or median (interquartile range). Data were analysed within the same sex and age using one-way analysis of variance (ANOVA) or non-parametric alternatives with Tukey's *post hoc* multiple comparisons where appropriate. Data were log-transformed prior to analysis if data followed a log-normal distribution. Statistical outliers were determined via the Grubb's method in Prism.

Myography dose–response curves were analysed using repeated measures mixed effects models. Longitudinal SBP and HR data were analysed using repeated measures two-way ANOVA (full dataset) or mixed effects models (random missing data) to assess the effect of age, the maternal environment, and the interaction between them. Correlations were determined using two-tailed Pearson correlation on normally distributed or log-transformed data, where applicable. $P < 0.05$ was considered statistically significant.

## Results

### Body composition

Detailed body composition and metabolic phenotyping was published previously (Schoonejans et al., 2022); there were no differences in body weight in male offspring at the time blood pressure and cardiac function was recorded, but female Ob-Met offspring had increased body weight at 12 months of age compared to offspring of Con and Ob dams.

### Arterial blood pressure and peripheral vascular function is altered in 12-month-old offspring of obese metformin-treated dams

**Female data.** Female Ob offspring had increased SBP compared to Con offspring at 12 months of age (Fig. 2*A*), an effect that was observed from 3 months of age (Table 3). Ob-Met offspring also had increased SBP compared to Con offspring at 12 months of age (Fig. 2*A* and Table 3). Female 12-month-old Ob-Met offspring also had decreased HR compared to both Con and Ob offspring from 3 months of age (Table 3). SBP decreased between 3 and 6 or 12 months of age across the cohort, most probably a result of enhanced habituation to the tail cuff technique (Table 3).

Femoral arteries from female offspring of obese dams showed increased reactivity to $K^+$, independent of the metformin intervention (Fig. 2*B*; see also Supporting information, Data S1), representing an increased constrictor reactivity in these vessels. Because increased

**Table 3. Systolic blood pressure and heart rate within the same animals across the lifecourse**

| | Female offspring SBP (mmHg) | | | Female offspring HR (beats min$^{-1}$) | | | Male offspring SBP (mmHg) | | | Male offspring HR (beats min$^{-1}$) | | |
|---|---|---|---|---|---|---|---|---|---|---|---|---|
| | 3 months | 6 months | 12 months | 3 months | 6 months | 12 months | 3 months | 6 months | 12 months | 3 months | 6 months | 12 months |
| **Con** | **108** | **101** | **99** | **638** | **645** | **652** | **112** | **106** | **109** | **520** | **573** | **614** |
| SD | 12 | 9 | 9 | 71 | 46 | 38 | 5 | 10 | 15 | 59 | 45 | 65 |
| SEM | 4 | 3 | 3 | 21 | 13 | 11 | 2 | 3 | 4 | 18 | 15 | 19 |
| *n* | *11* | *11* | *12* | *11* | *12* | *12* | *9* | *12* | *11* | *11* | *9* | *11* |
| **Ob** | **119** | **106** | **108** | **631** | **670** | **659** | **110** | **110** | **105** | **554** | **570** | **593** |
| SD | 16 | 12 | 8 | 60 | 28 | 28 | 12 | 16 | 9 | 50 | 30 | 53 |
| SEM | 5 | 4 | 2 | 18 | 9 | 8 | 3 | 5 | 3 | 15 | 9 | 16 |
| *n* | *11* | *12* | *12* | *11* | *10* | *11* | *13* | *10* | *11* | *12* | *11* | *11* |
| **Ob-Met** | **110** | **102** | **111** | **611** | **620** | **609** | **111** | **102** | **107** | **588** | **593** | **586** |
| SD | 9 | 8 | 10 | 69 | 37 | 42 | 15 | 7 | 10 | 50 | 60 | 69 |
| SEM | 3 | 2 | 3 | 20 | 11 | 12 | 4 | 2 | 3 | 17 | 18 | 20 |
| *n* | *12* | *12* | *12* | *12* | *11* | *12* | *12* | *11* | *12* | *9* | *11* | *12* |
| **Mixed model** | | | | | | | | | | | | |
| | Age | Group | A * G | Age | Group | A * G | Age | Group | A * G | Age | Group | A * G |
| P value | **0.0015** | **0.0154** | 0.2307 | 0.2468 | **0.0458** | 0.6726 | 0.1645 | 0.6532 | 0.4908 | **0.0009** | 0.5961 | **0.0252** |
| Age effect | 6 *vs.* 3m | 12 *vs.* 3m | 12 *vs.* 6m | 6 *vs.* 3m | 12 *vs.* 3m | 12 *vs.* 6m | 6 *vs.* 3m | 12 *vs.* 3m | 12 *vs.* 6m | 6 *vs.* 3m | 12 *vs.* 3m | 12 *vs.* 6m |
| Post hoc P | **0.0054** | **0.0316** | 0.5155 | 0.2589 | 0.5958 | 0.6379 | 0.16 | 0.2212 | 0.9029 | 0.0519 | **0.0029** | 0.2851 |
| Group effect | Ob *vs.* Con | Ob-Met *vs.* Con | Ob-Met *vs.* Ob | Ob *vs.* Con | Ob-Met *vs.* Con | Ob-Met *vs.* Ob | Ob *vs.* Con | Ob-Met *vs.* Con | Ob-Met *vs.* Ob | Ob *vs.* Con | Ob-Met *vs.* Con | Ob-Met *vs.* Ob |
| Post hoc P | **0.0190** | 0.1169 | 0.4761 | 0.9252 | **0.0266** | **0.0072** | 0.9926 | 0.703 | 0.7976 | 0.9792 | 0.4357 | 0.4018 |
| Age effect/group | 6 *vs.* 3m | 12 *vs.* 3m | 12 *vs.* 6m | 6 *vs.* 3m | 12 *vs.* 3m | 12 *vs.* 6m | 6 *vs.* 3m | 12 *vs.* 3m | 12 *vs.* 6m | 6 *vs.* 3m | 12 *vs.* 3m | 12 *vs.* 6m |
| Con P | 0.3083 | 0.9422 | 0.9249 | 0.9342 | 0.6803 | 0.9285 | 0.1064 | 0.8083 | 0.8695 | 0.0843 | **0.0040** | 0.1209 |
| Ob P | 0.1200 | 0.0521 | 0.9444 | 0.1115 | 0.5393 | 0.0825 | 0.9983 | 0.2943 | 0.6475 | 0.3091 | **0.0484** | 0.1868 |
| Ob-Met P | 0.1207 | 0.9725 | **0.0118** | 0.9005 | 0.9943 | 0.7552 | 0.2153 | 0.6452 | 0.3736 | 0.9778 | 0.9972 | 0.9717 |
| Group effect/age | Ob *vs.* Con | Ob-Met *vs.* Con | Ob-Met *vs.* Ob | Ob *vs.* Con | Ob-Met *vs.* Con | Ob-Met *vs.* Ob | Ob *vs.* Con | Ob-Met *vs.* Con | Ob-Met *vs.* Ob | Ob *vs.* Con | Ob-Met *vs.* Con | Ob-Met *vs.* Ob |
| 3 months P | 0.2046 | 0.9422 | 0.2355 | 0.9664 | 0.6273 | 0.7348 | 0.8399 | 0.9740 | 0.9797 | 0.3366 | **0.0322** | 0.2936 |
| 6 months P | 0.4651 | 0.9181 | 0.6358 | 0.2983 | 0.3278 | **0.0064** | 0.7600 | 0.4148 | 0.2858 | 0.9923 | 0.6774 | 0.5319 |
| 12 months P | 0.0657 | **0.0209** | 0.7313 | 0.9947 | **0.0377** | 0.0552 | 0.7095 | 0.911 | 0.869 | 0.6685 | 0.5828 | 0.967 |

*P* values reflect repeated measures mixed effects modelling with Tukey's multiple comparisons. A * G, interaction term between Age and Group; Con, offspring of control-fed dams; HR, heart rate; Ob, offspring of obesogenic diet-fed dams; Ob-Met, offspring of dams fed an obesogenic diet supplemented with metformin in gestation; SBP, systolic blood pressure. Blood pressure and heart rate measurements were performed in the same animals across the three timepoints. Statistically significant comparisons (*P* < 0.05) are highlighted in bold text.

vascular resistance in the peripheral circulation can affect the reflected blood pressure waveform towards the heart and lead to main outflow tract wall remodelling, the thickness of the tunica media of the aorta was measured. Aortic tunica media thickness was significantly increased in Ob offspring, and this was not corrected by the metformin intervention (Fig. 2*D*). Aortic tunica media thickness correlated positively with the femoral $K^+$ response (Table 4). No difference in femoral vasoconstrictor response to PE, SNP or ACh was found between groups (Fig. 2*C*; see also Supporting information, Data S1).

SBP at 12 months of age was positively correlated with measures of adiposity, serum insulin, serum leptin, and the femoral vascular response to $K^+$ (Table 4).

**Male data.** By contrast to the female data, there was no difference in SBP between groups in male offspring at any age (Fig. 2*E* and Table 3). Decreased HR was observed in male Ob-Met offspring at 3 months of age (Table 3), but this did not persist. There was no difference in maximal contractile response (reactivity to $K^+$) in femoral arteries of 12-month-old male offspring (Fig. 2*F*; see also Supporting information, Data S1), but male Ob-Met offspring showed increased femoral constrictor reactivity to PE (Fig. 2*G*; see also Supporting information, Data S1). There was no difference in vasodilatory response to ACh or SNP (see Supporting information, Data S1) or in aortic tunica media thickness (see Supporting information, Data S1) in 12-month-old male offspring.

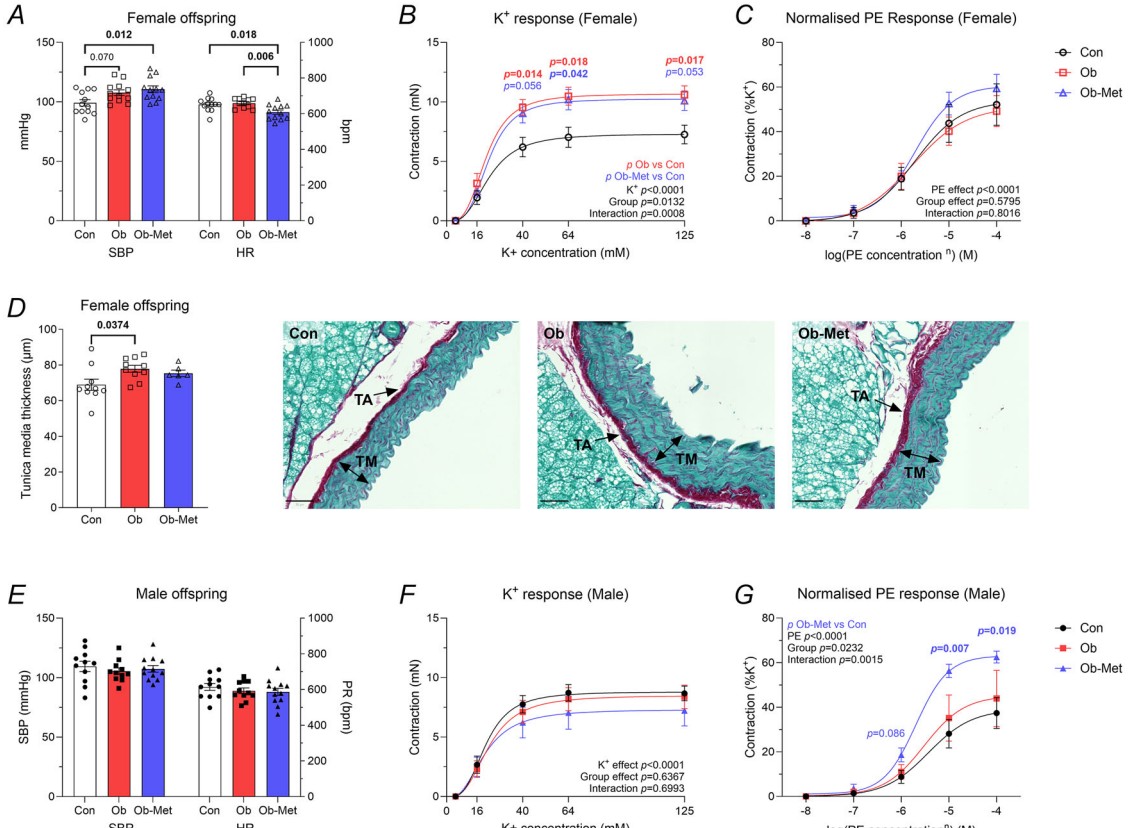

**Figure 2. Blood pressure and vascular function in 12-month-old offspring**

*A*, systolic blood pressure (SBP) and heart rate (HR) measured in female offspring by tail cuff plethysmography. *B–C*, femoral arterial response curves to (*B*) increasing concentrations of $K^+$ ($n = 11$ Con, $n = 9$ Ob, $n = 11$ Ob-Met), or (*C*) increasing doses of phenylephrine (PE) normalised to $10^{-8}$ M PE and the vessel's maximal contraction to elevated $K^+$ ($n = 11$ Con, $n = 9$ Ob, $n = 11$ Ob-Met), measured in female offspring using wire myography. *D*, aortic tunica media thickness in 12-month-old female offspring as measured by histology and imaged using brightfield light microscopy ($n = 10$ Con, $n = 10$ Ob, $n = 6$ Ob-Met) with representative images of aortas from Con, Ob and Ob-Met females. Scale bar (lower left corner) = 50 μm. Arrows point to tunica media (TM) and tunica adventitia (TA). *E*, SBP and HR measured in male offspring by tail cuff plethysmography. *F* and *G*, femoral arterial response curves to (*F*) increasing concentrations of $K^+$ ($n = 9$ Con, $n = 9$ Ob, $n = 7$ Ob-Met), or (*G*) increasing doses of PE normalised to $10^{-8}$ M PE and the vessel's maximal contraction to elevated $K^+$ ($n = 8$ Con, $n = 5$ Ob, $n = 7$ Ob-Met), measured in male offspring using wire myography. *P* values reflect one-way ANOVA with Tukey's multiple comparisons (*A*, *D* and *E*) or using repeated measures mixed model analysis with Tukey's multiple comparisons (*B*, *C*, *F* and *G*). Significant *P* values are shown in bold. Curves are the best non-linear sigmoidal curve fit.

**Table 4. Correlations with systolic blood pressure and myography data in 12-month-old female offspring**

|  | Parameter | Pearson *r* | Pearson *P* | *n* |
|---|---|---|---|---|
| SBP at 3 months | IVRT at 3 months | 0.7532 | 0.0002 | 19 |
| SBP at 12 months | Total WAT collected (g) | 0.4782 | 0.0056 | 32 |
|  | Total WAT collected (% of body weight) | 0.4773 | 0.0057 | 32 |
|  | Serum insulin (pmol L$^{-1}$)[a,b,c] | 0.3867 | 0.0288 | 32 |
|  | Serum leptin (ng mL$^{-1}$)[a,b,c] | 0.4617 | 0.0060 | 34 |
|  | Femoral K$^+$ response (AUC)[b] | 0.3852 | 0.0324 | 31 |
| K$^+$ AUC at 12 months | Tunica media thickness (μm)[b] | 0.5967 | 0.0043 | 21 |

*P* values reflect two-tailed Pearson correlations.
[a] Correlation performed on log-transformed data.
[b] Correlation performed using littermate data.
[c] Serum data as previously published (Schoonejans et al., 2022). AUC, area under the curve; IVRT, isovolumetric relaxation time; SBP, systolic blood pressure; WAT, white adipose tissue.

### Early left ventricular cardiac abnormalities in female offspring of obese and obese metformin-treated dams do not persist until 12 months

Key findings from echocardiography studies are highlighted in the text. The full dataset for female and male offspring at 3, 6 and 12 months is included in the Supporting information (Data S2–S7).

At 3 months of age, female Ob offspring showed increased isovolumetric relaxation time (IVRT, both in absolute time and proportion of the cardiac cycle), decreased deceleration time (DT) and an increase in the myocardial performance index (MPI) compared to Con offspring (Fig. 3*A*–*C*; see also Supporting information, Data S2). Female Ob-Met offspring did not show these changes but exhibited a decrease in the ratio between the E-wave (early ventricular filling) and A-wave (atrial contraction-related ventricular filling) velocities (E/A ratio) at 3 months of age (Fig. 3*D*). IVRT in female offspring correlated positively to SBP at 3 months (Table 4). All indices of diastolic function normalised by 6 months. At 6 months of age, female Ob-Met offspring showed decreased stroke volume (SV) and cardiac output (CO) compared to both Con and Ob offspring, related to a fall in end-diastolic LV volume and area (Fig. 3*E*–*G*; see also Supporting information, Data S3). There was no significant difference in LV morphology or systolic and diastolic function between groups in 12-month-old female offspring (see Supporting information, Data S4). Cardiac weight may be slightly increased in female Ob-Met offspring, but this did not reach statistical significance (Fig. 3*H*). There was no difference in LV collagen mRNA expression between female offspring (see Supporting information, Data S1).

### Male offspring of obese and obese metformin-treated dams have diastolic dysfunction

Male Ob-Met offspring had increased CO at 3 months compared to Ob offspring (Fig. 4*A*; see also Supporting information, Data S5). By 6 months, however, CO and SV were decreased in Ob-Met compared to Ob offspring (Fig. 4*A* and *B*; see also Supporting information, Data S6), but this did not persist until 12 months of age (Fig. 4*A* and *B*; see also Supporting information, Data S7). At 12 months of age, male Ob offspring showed differences in diastolic function compared to Con offspring: decreased A-wave velocity resulting in an increase in the E/A ratio (Fig. 4*D* and *E*). This change in A-wave velocity was not present at 6 months of age although the E/A ratio was already increased (Fig. 4*E*), indicative of progressive diastolic dysfunction. Metformin treatment did not prevent the decrease in A-wave velocity: in Ob-Met offspring this change occurred earlier and was apparent at 6 months of age (Fig. 4*D*), suggesting earlier development of restrictive ventricular filling. Moreover, Ob-Met offspring had decreased DT compared to both Con and Ob offspring at 6 months of age, also consistent with restrictive ventricular filling (Fig. 4*F*). The decrease in A-wave velocity and DT persisted until 12 months of age (Fig. 4*D* and *F*), although the E/A ratio was not significantly different from either Con or Ob offspring. To investigate potential cardiac fibrosis, collagen levels were assessed and the expression of *Col4a1* mRNA was increased in Ob-Met offspring compared to Con offspring (Fig. 4*G*). *Col4a1* expression was inversely correlated with DT at 12 months (Fig. 4*H*). There was no significant difference in LV internal dimensions at 12 months of age (see Supporting information, Data S7), but cardiac weight was increased in 12-month-old male Ob-Met offspring compared to controls (Fig. 4*I*).

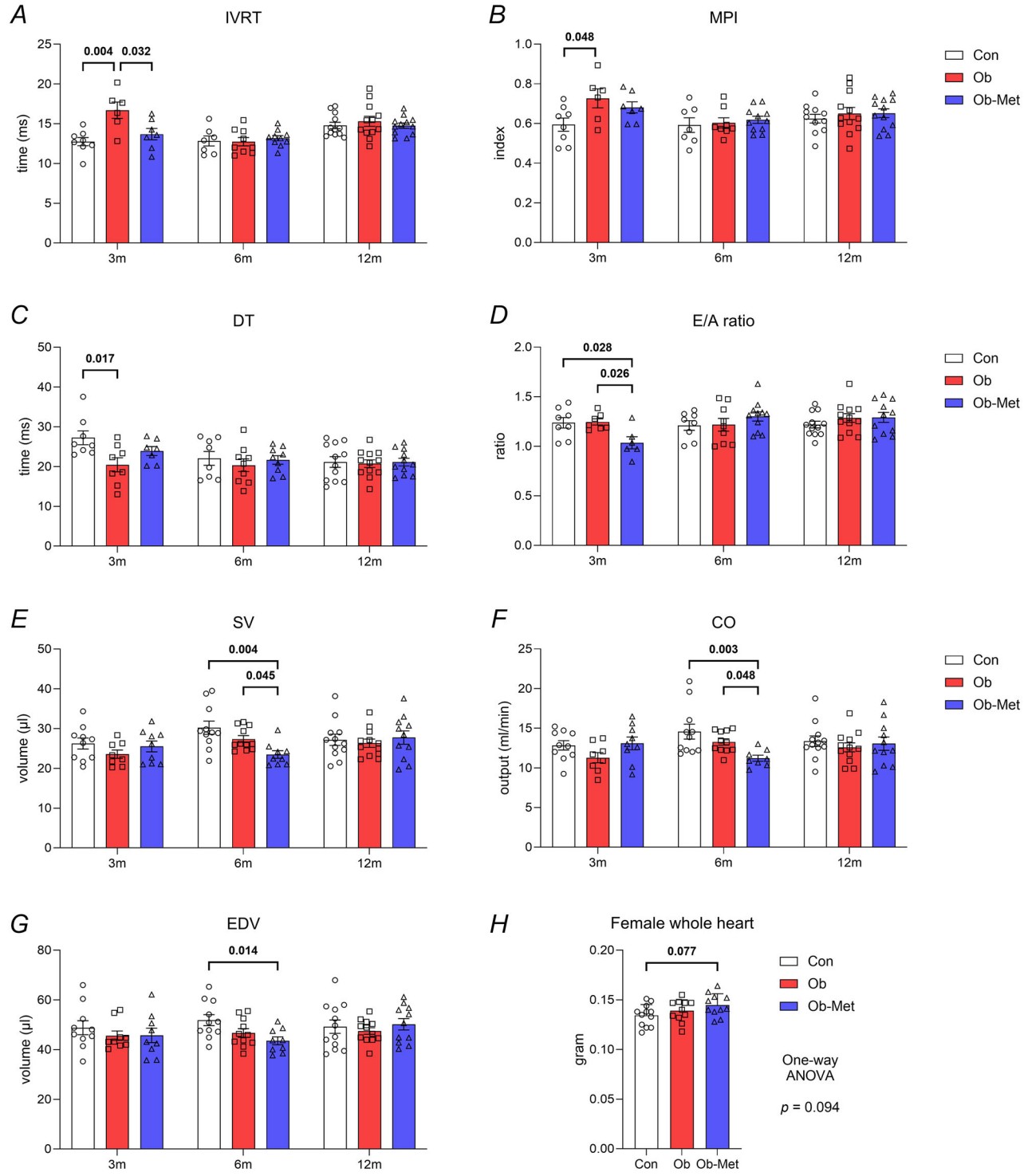

**Figure 3. Left ventricular function in female offspring**

*A*, isovolumetric relaxation time (IVRT, *n* = 7–12 per group). *B*, left ventricular myocardial performance index (MPI, *n* = 6–12 per group). *C*, deceleration time (DT, *n* = 7–12 per group). *D*, ratio between early and atrial contraction-related ventricular filling velocity (E/A, *n* = 6–12 per group). *E*, stroke volume (SV, *n* = 8–12 per group). *F*, cardiac output (CO, *n* = 8–12 per group). *G*, end-diastolic left ventricular volume (EDV, *n* = 9–12 per group) in the same 3-, 6- and 12-month-old female offspring using echocardiography. *H*, cardiac weight as determined by dissection at 12 months of age (*n* = 11–12 per group). *P* values reflect one-way ANOVA with Tukey's multiple comparisons, or non-parametric alternative where appropriate. Significant *P* values are shown in bold.

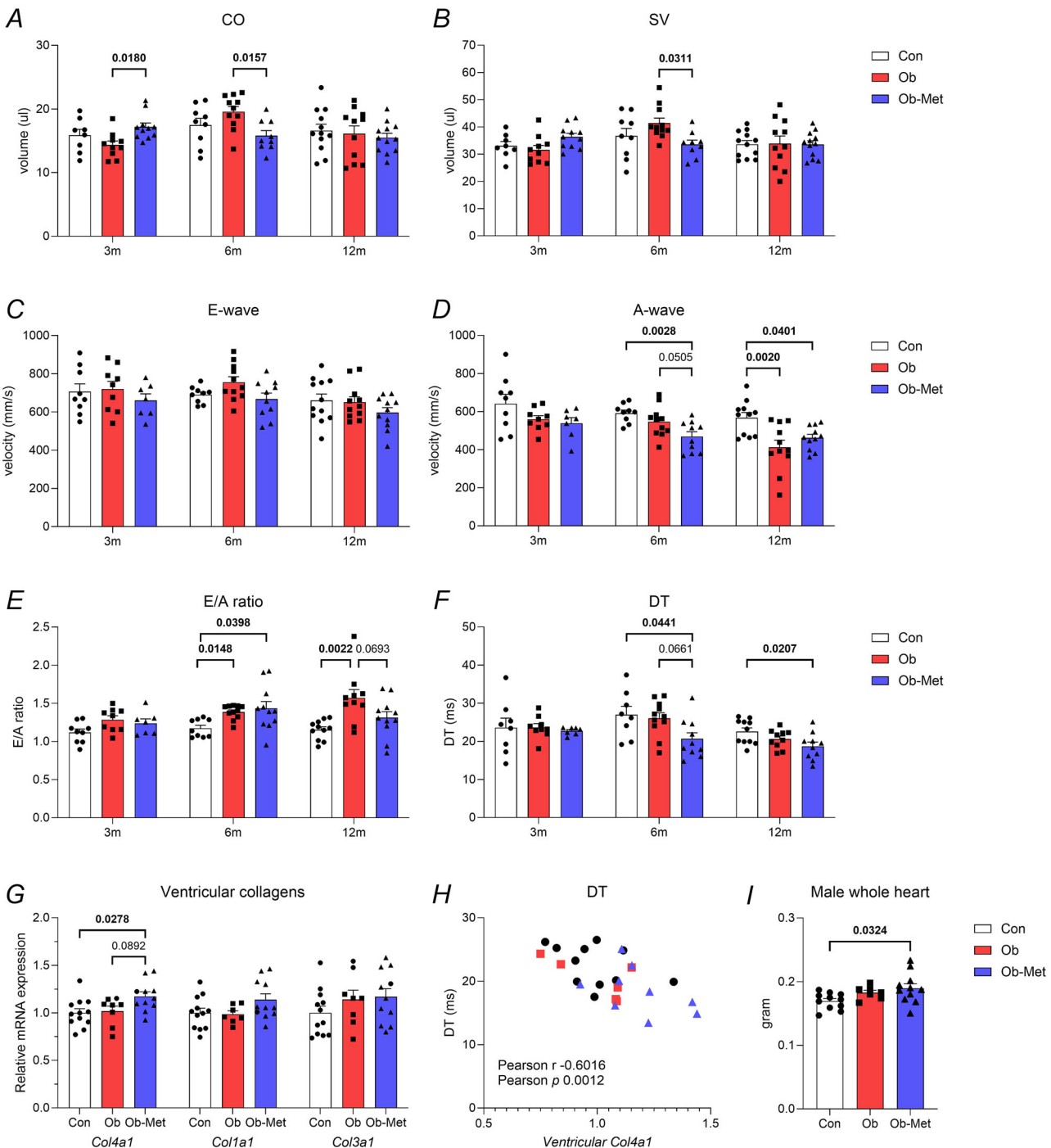

**Figure 4. Left ventricular function in male offspring**
*A*, cardiac output (CO, *n* = 8–12 per group). *B*, stroke volume (SV, *n* = 8–12 per group). *C*, early ventricular filling velocity (E-wave, *n* = 7–12 per group). *D*, atrial contraction-related ventricular filling velocity (A-wave, *n* = 7–12 per group). *E*, ratio between early and atrial contraction-related ventricular filling velocity (E/A, *n* = 7–12 per group). *F*, deceleration time (DT, *n* = 7–12 per group) in 3-, 6- and 12-month-old male offspring using echocardiography. *G*, expression of left ventricular collagen genes by real time quantitative PCR (*n* = 7–12 per group). *H*, correlation between mitral valve deceleration time (DT) and left ventricular expression of *Col4a1*, measured in sibling pairs (*n* = 26). *P* values reflect one-way ANOVA with Tukey's multiple comparisons or non-parametric alternative where appropriate (*A–G* and *I*); or two-tailed Pearson correlation analysis (*H*). Significant *P* values are shown in bold.

## Discussion

The prevalence of diabetes in pregnancy is rapidly increasing, as is the use of metformin in its treatment (Cesta et al., 2019). A substantial proportion of the new generation will thus have been exposed to metformin *in utero*, yet it will take years before long-term follow-up data in humans are available. The present study in mice identified long-term cardiovascular effects of both maternal obesity and prenatal metformin exposure on aged offspring, with clear sex-specific effects (Fig. 5), and investigated mechanisms at the organ level.

The data obtained in this study show that maternal obesity during pregnancy led to an increase in SBP in female mouse offspring throughout life. At 12 months of age, female offspring of obese dams also showed an increase in femoral artery contractile reactivity and aortic wall remodelling, whereas abnormalities in cardiac

diastolic function seen in earlier adulthood were no longer detectable. By contrast, in male offspring, maternal obesity led to diastolic dysfunction as early as 6 months of age without changes in systolic function, vascular reactivity, or blood pressure. In male or female offspring, the metformin intervention did not correct these phenotypes. In females, metformin exposure appeared to delay the increase in SBP and cause a fall in HR throughout life, consistent with enhanced and sustained activation of cardiac baroreflex function. Furthermore, in females, metformin exposure did not improve vascular abnormalities, nor prevent early life changes in diastolic function. Male metformin-exposed offspring not only had similar indices of diastolic dysfunction to offspring of untreated dams, but also had additional cardiomegaly, cardiac collagen expression and vascular sympathetic hyper-reactivity, suggesting that metformin exposure resulted in a worse cardiovascular phenotype.

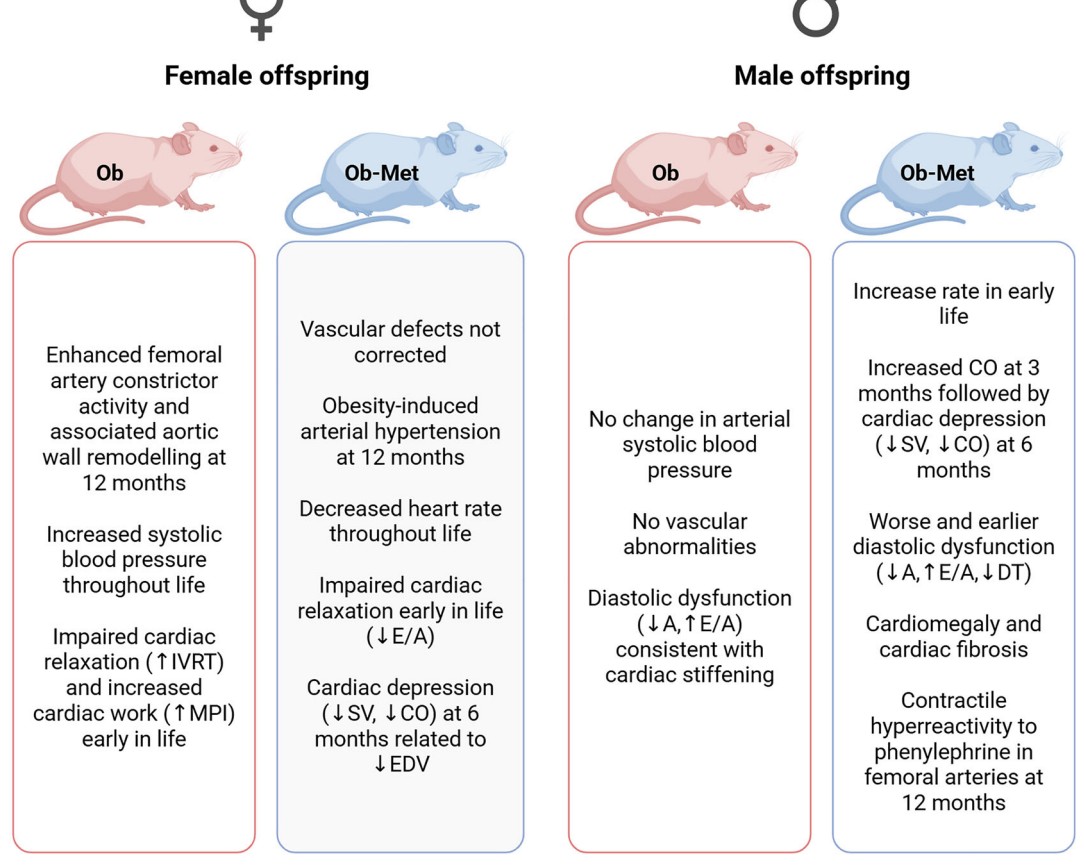

**Figure 5. Summary of cardiovascular outcomes in aged offspring**
Female offspring (left) and male offspring (right). Ob (red), offspring of obesogenic diet-fed dams; Ob-Met (blue), offspring of dams fed an obesogenic diet supplemented with metformin in gestation. Created with BioRender.com.

## Cardiovascular phenotype of female offspring

Sustained increases in peripheral vascular resistance can impact on aortic wall remodelling because the enhanced speed of the reflected pressure waveform propagating within the arterial tree can increase cardiac and main outflow tract load, as occurs in ageing patients and those with peripheral vascular stiffness and disease (McEniery et al., 2007). Therefore, it was of interest to determine whether the increased peripheral vascular constrictor phenotype seen in female Ob offspring in the present study also led to changes in aortic wall thickness. The findings revealed that female Ob offspring had increased SBP from 3 months until 12 months of age, consistent with previous mouse reports up to 6 months of age (Samuelsson et al., 2008), and increased aorta tunica media thickness at 12 months of age. Both SBP and aortic tunica media thickness correlated with the degree of the femoral $K^+$ vasoconstrictor response, supporting that programmed changes in peripheral vascular function may contribute to both hypertension and aortic wall remodelling. The metformin intervention did not ameliorate these vascular phenotypes and Ob-Met females developed obesity-induced hypertension by 12 months of age. No differences in femoral vascular reactivity to alpha-adrenergic agonists (PE) or to smooth muscle-dependent (SNP) and endothelium-dependent (ACh) dilator reactivity in the femoral circulation were found between groups. The SBP correlated with measures of adiposity, hyperleptinaemia and hyperinsulinaemia (Schoonejans et al., 2022), all of which may have contributed to the rise in blood pressure, but leptin specifically has been implicated in the development of adiposity-related hypertension (Samuelsson et al., 2013). Therefore, the increased adiposity and metabolic adverse effects of intrauterine metformin exposure appear to worsen outcomes in Ob-Met female offspring in the longer term, probably overwhelming any compensatory baroreflex fall in basal HR in response to sustained elevations in basal SBP.

Female Ob and Ob-Met offspring showed signs of impaired LV relaxation (changes IVRT/DT and E/A, respectively) as well as increased cardiac work (increased MPI) in early life. IVRT correlated positively with SBP at 3 months, suggesting that diastolic abnormalities in Ob offspring may be secondary to hypertension. Between 3 and 6 months, these differences disappeared. The decrease in SV and CO in 6-month-old Ob-Met offspring was also restored with age because no differences in LV systolic function were observed between groups at 12 months. Female offspring may thus successfully compensate for the increased afterload associated with hypertension until at least 1 year of age, potentially related to the protective effects of oestrogen, which have been shown to offset programmed cardiac defects in offspring

of high fat-fed dams (Chen et al., 2021). Nevertheless, we have previously reported that maternal and post-natal obesogenic diet have additive effects on arterial blood pressure, promoting hypertension in 8-week-old male offspring (Loche et al., 2018). Therefore, female Ob/Ob-Met offspring could be susceptible to developing severe hypertension and associated CVD risk, if exposed to a calorie-rich environment.

A study in mice found changes in E/A in female offspring of obese dams between 9 and 24 months of age (Vaughan et al., 2022). Although the sample size in this previous study was considerably smaller ($n = 2–6$ per group) than ours, the 12-month-old mice in our study may have been studied during the 'pseudonormalisation' stage of diastolic dysfunction, indicating a delay rather than prevention of a cardiac dysfunctional phenotype, potentially related to the oestropause-associated fall in oestrogen, which, in mice, occurs between 9 and 18 months of life (Aiken & Ozanne, 2013).

## Cardiovascular phenotype of male offspring

Male Ob offspring showed a decrease in the A-wave velocity and accompanying increased E/A ratio at 12 months of age. These effects represent signs of restrictive diastolic filling with progressive diastolic dysfunction and cardiac stiffening (Lalande & Johnson, 2008). This may be a directly programmed phenotype because we have previously reported the histological presence of cardiac fibrosis at 8 weeks and increased *ex vivo* end-diastolic filling pressures at 12 weeks of age in this mouse model of maternal obesity during pregnancy (Blackmore et al., 2014; Loche et al., 2018). Others have also found diastolic dysfunction with ventricular fibrosis in 3- and 6-month-old mouse offspring of HFHS-fed dams (Vaughan et al., 2020), consistent with our findings. Intrauterine hypoxia, as demonstrated in our model (Fernandez-Twinn et al., 2017), also promotes cardiac stiffness in rat offspring (Niu et al., 2018).

The metformin intervention did not correct diastolic dysfunction in male offspring. The intervention was associated with a worse and earlier-onset fall in the A-wave velocity from 6 months of age, indicative greater restrictive filling in Ob-Met *vs.* Ob male offspring. The latter is also supported by the concurrent decrease in DT not seen in Ob offspring. Together with the increased heart weight and enhanced cardiac *Col4a1* mRNA expression in male 12-month-old Ob-Met offspring, which also correlated to decreased DT, this suggests that male Ob-Met hearts are even stiffer than those in Ob offspring. The metformin intervention was also associated with independent adverse cardiovascular effects not seen in Ob offspring, such as the alpha-adrenergic hyperactivity of femoral arteries at 12 months and the changes in CO at

3 and 6 months of age. Because both male and female Ob-Met showed a fall in SV and CO at 6 months of age, and because maternal adiposity and metabolic health are improved with metformin treatment in our model (Hufnagel et al., 2022; Schoonejans et al., 2021), this raises the question whether a direct adverse intrauterine effect on the offspring cardiovascular system may have occurred, promoting this phenotype of depressed cardiac function in offspring of both sexes.

## Our findings in context of existing animal models and human studies

Animal studies addressing cardiovascular health in metformin-exposed offspring are limited. Studies in chow-fed pregnancy found no intrinsic effect of prenatal metformin on 11-week-old offspring aortic responses to PE, Ach and SNP (Novi et al., 2017; Vidigal et al., 2018). By contrast, prenatal metformin exposure prevented hypertension programmed by high-fructose-feeding in 12-week-old male rats, but it was also found that metformin impaired nitric oxide homeostasis, which may negatively impact vascular function later in life (Tain et al., 2018). The offspring in these reports were studied at a relatively young age. This is noteworthy because cardiovascular dysfunctional phenotypes often emerge later in life and our data suggest that metformin may lead to an accelerated ageing phenotype because Ob-Met males show more severe age-related cardiovascular and adipose tissue dysfunction (Schoonejans et al., 2022).

Cardiac function in humans has (to our knowledge) only been studied in 4-year-old offspring of obese pregnancies treated with metformin compared to placebo, where changes in diastolic parameters were observed that were considered of putative benefit (Panagiotopoulou et al., 2020). This human study was not powered to detect sex differences. A pilot study of children born to women with PCOS showed SBP was increased in 8-year-olds born to metformin-treated pregnant PCOS individuals (Rø et al., 2012). Although this study was small, this finding is of interest considering the vascular abnormalities observed in our Ob-Met mice. Follow-up studies in PCOS do not replicate this finding but did describe a borderline increase in a composite outcome of 'metabolically abnormal obese' in 5–10-year-old metformin-exposed children including hypertension (Hanem et al., 2019). Although this finding was not sex-stratified because of a lack of power, Hanem et al. (2019) did sex-segregate most other data and reported increased HR in metformin-exposed boys not dissimilar to the elevation seen in 3-month-old male Ob-Met offspring in the present study. Other studies in GDM or obese pregnancies found no differences in cardio-vascular parameters in young children (2 and 9 years, respectively) following *in utero* metformin exposure (Rowan et al., 2011; Yang et al., 2022). Our work has shown that impairments may not appear until later in life: follow up of these children until older age is therefore crucial.

The main strength of the present study is the longitudinal follow up of male and female offspring, allowing investigation into the development of sex-specific cardiovascular phenotypes within each animal across the life course. To our knowledge, cardiac function has not previously been assessed in adult offspring and no vascular functional phenotyping has been carried out beyond young adult life (rodents) or childhood (children). As far as we are aware, cardio-vascular responses to *in utero* metformin exposure have not been studied before in female offspring. However, the longitudinal nature of the work also introduces a limitation because functional experiments were prioritised over the use of tissues for molecular or histological analysis. Therefore, no tissues were available for *ex vivo* analysis at 3 or 6 months of age. Our model of glucose intolerance first detected in pregnancy is clinically relevant to GDM. Although the timing of metformin treatment may be more similar to that for women with pre-gestational diabetes or PCOS, the similarity in serum metformin concentrations between our dams and pregnant women (Hufnagel et al., 2022) strengthens the need to look for parallels between our data and human studies.

## Conclusions

Maternal obesity leads to sex-dependent cardiovascular abnormalities in aged offspring. Metformin intervention during obese glucose intolerant pregnancy did not correct this and introduced further unique cardio-vascular alterations in both sexes. Female offspring of obese dams had elevated blood pressure throughout life, cardiac diastolic dysfunction at 3 months, and increased femoral vasoconstrictor reactivity and aortic wall remodelling at 12 months. Metformin treatment did not ameliorate this and led to obesity-induced hypertension at 12 months. Male offspring of obese pregnancy irrespective of metformin had cardiac diastolic dysfunction from 6 months without changes in vascular reactivity or blood pressure. Male metformin-exposed offspring also showed cardiomegaly, increased cardiac collagen and vascular sympathetic hyper-reactivity, suggesting metformin exposure worsened the cardio-vascular phenotype. The data highlight that long-term cardiovascular follow up in offspring of both sexes from human pregnancies treated with metformin is crucial.

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

## Additional information

### Data availability statement

The data underlying this article are available in the article and from the corresponding author upon reasonable request.

### Competing interests

The authors declare that there are no relationships or activities that might bias, or be perceived to bias, their work.

### Author contributions

J.M.S: data curation, formal analysis, investigation, project administration, visualisation, writing – original draft preparation, writing – review & editing. P.W: data curation,

formal analysis, investigation, visualisation, writing – review & editing. L.V.M: data curation, investigation, writing – review & editing. K.K.W: investigation, validation. T.J.A: investigation. T.G: methodology. H.L.B: methodology, validation. O.V.P: writing – review & editing. D.S.F.T: investigation, methodology. D.A.G: methodology, supervision, data interpretation, writing – review & editing. S.E.O: conceptualisation, funding acquisition, methodology, project administration, resources, supervision, writing – review & editing. All authors have approved the final version of the manuscript submitted for publication. All authors agree to be accountable for all aspects of the work in ensuring that questions related to the accuracy or integrity of any part of the work are appropriately investigated and resolved. All persons designated as authors qualify for authorship, and all those who qualify for authorship are listed.

## Funding

This work was funded by the British Heart Foundation [grants RG/17/12/33167, PG/13/46/30329, PG/20/11/34957 and studentship to JMS (FS/16/53/32729)], the Wellcome Trust [PhD studentship to PW (215242/Z/19/Z)] and the Medical Research Council (MC_UU_00014/4). The IMS-MRL Histopathology Core is funded by the Medical Research Council (MRC_MC_ UU_00014/5) and the IMS-MRL Imaging core is funded by a Wellcome Major Award (208363/Z/17/Z).

## Acknowledgements

We thank C. Custance (Wellcome-MRC Institute of Metabolic Science-Metabolic Research Laboratories, University of Cambridge, Cambridge, UK) for expert technical assistance. We also thank the Histopathology Core for the use of their processing facilities (MRC Metabolic Diseases Unit, University of Cambridge).

## Keywords

cardiac function, gestational diabetes mellitus, hypertension, metformin

## Supporting information

Additional supporting information can be found online in the Supporting Information section at the end of the HTML view of the article. Supporting information files available:

**Peer Review History**
**Supporting Information, Data S1**
**Supporting Information, Data S2**
**Supporting Information, Data S3**
**Supporting Information, Data S4**
**Supporting Information, Data S5**
**Supporting Information, Data S6**
**Supporting Information, Data S7**

