## [Peer Review History · The Journal of Physiology]

Cardiovascular outcome in 12-month-old male and female offspring of metformin-treated obese mice

Josca Mari ette Schoonejans, Phoebe Wilsmore, Lais Vales Mennitti, Kwun Kiu Wong, Thomas J Ashmore, Tessa A.C. Garrud, Heather Louise Blackmore, Olga Patey, Denise S Fernandez-Twinn, Dino A Giussani, and Susan E Ozanne
DOI: 10.1113/JP288696

Corresponding author(s): Josca Schoonejans (josca.schoonejans11@imperial.ac.uk)

The following individual(s) involved in review of this submission have agreed to reveal their identity: Karen Forbes (Referee #2)

Review Timeline:

Submission Date:	06-Feb-2025
Editorial Decision:	02-Apr-2025
Revision Received:	29-Apr-2025
Accepted:	24-May-2025

Senior Editor: Laura Bennet

Reviewing Editor: Christopher Lear

Transaction Report:

Dear Dr Schoonejans,

Re: JP-RP-2025-288696 "**Cardiovascular outcome in 12-month-old male and female offspring of metformin-treated obese mice**" by Josca Mariëtte Schoonejans, Phoebe Wilsmore, Lais Vales Mennitti, Kwun Kiu Wong, Thomas J Ashmore, Tessa A.C. Garrud, Heather Louise Blackmore, Olga Patey, Denise S Fernandez-Twinn, Dino A Giussani, and Susan E Ozanne

Thank you for submitting your manuscript to The Journal of Physiology. It has been assessed by a Reviewing Editor and by 2 expert referees and we are pleased to tell you that it is acceptable for publication following satisfactory revision.

REVISION CHECKLIST:

We look forward to receiving your revised submission.

Yours sincerely,

Laura Bennet
Senior Editor
The Journal of Physiology

REQUIRED ITEMS FOR REVISION

- Include a Key Points list in the article itself, before the Abstract.
- Author photo and profile. First or joint first authors are asked to provide a short biography (no more than 100 words for one author or 150 words in total for joint first authors) and a portrait photograph. These should be uploaded and clearly labelled together in a Word document with the revised version of the manuscript. See Information for Authors for further details.
- The reference list must be in alphabetical order, rather than numbered, to comply with our Journal format.
- The Journal of Physiology funds authors of provisionally accepted papers to use the premium BioRender site to create high resolution schematic figures. Follow this link and enter your details and the manuscript number to create and download figures. Upload these as the figure files for your revised submission. If you choose not to take up this offer, we require figures to be of similar quality and resolution. If you are opting out of this service to authors, state this in the Comments section on the Detailed Information page of the submission form. The link provided should only be used for the purposes of this submission. Authors will be charged for figures created on this premium BioRender account if they are not related to this manuscript submission.
- Please upload separate high-quality figure files via the submission form.
- Your paper contains Supporting Information of a type that we no longer publish, including supplementary tables and figures. Any information essential to an understanding of the paper must be included as part of the main manuscript and figures. The only Supporting Information that we publish are video and audio, 3D structures, program codes and large data files. Your revised paper will be returned to you if it does not adhere to our Supporting Information Guidelines.
- Papers must comply with the Statistics Policy: https://jp.msubmit.net/cgi-bin/main.plex?form_type=display_requirements#statistics.

In summary:

- If $n \leq 30$, all data points must be plotted in the figure in a way that reveals their range and distribution. A bar graph with data points overlaid, a box and whisker plot or a violin plot (preferably with data points included) are acceptable formats.
- If $n > 30$, then the entire raw dataset must be made available either as supporting information, or hosted on a not-for-profit

repository, e.g. FigShare, with access details provided in the manuscript.

- 'n' clearly defined (e.g. x cells from y slices in z animals) in the Methods. Authors should be mindful of pseudoreplication.
- All relevant 'n' values must be clearly stated in the main text, figures and tables.
- The most appropriate summary statistic (e.g. mean or median and standard deviation) must be used. Standard Error of the Mean (SEM) alone is not permitted.
- Exact p values must be stated. Authors must not use 'greater than' or 'less than'. Exact p values must be stated to three significant figures even when 'no statistical significance' is claimed.

- Please include an Abstract Figure file, as well as the Figure Legend text within the main article file. The Abstract Figure is a piece of artwork designed to give readers an immediate understanding of the research and should summarise the main conclusions. If possible, the image should be easily 'readable' from left to right or top to bottom. It should show the physiological relevance of the manuscript so readers can assess the importance and content of its findings. Abstract Figures should not merely recapitulate other figures in the manuscript. Please try to keep the diagram as simple as possible and without superfluous information that may distract from the main conclusion(s). Abstract Figures must be provided by authors no later than the revised manuscript stage and should be uploaded as a separate file during online submission labelled as File Type 'Abstract Figure'. Please also ensure that you include the figure legend in the main article file. All Abstract Figures should be created using BioRender. Authors should use The Journal's premium BioRender account to export high-resolution images. Details on how to use and access the premium account are included as part of this email.

EDITOR COMMENTS

Reviewing Editor:

Thank you for submitting your well written manuscript detailing the effects of antenatal metformin on offspring outcomes. The reviewers have generally been positive about the importance of this topic and the methodology utilised. Please address their queries and concerns.

REFEREE COMMENTS

Referee #1:

Schoonejans et al present a study which examines the impact of obese pregnancy plus or minus metformin exposure on 12 month old male and female offspring. This is a well written and nicely presented study. It is topical and will be of interest to the readers of this journal. I only really had a few minor queries.

Intro: Well written intro. I just have one query. You describe human studies that you have outlined showing subtle cardiovascular effects in children who were exposed in utero to metformin. It is clear that they are impacted by the metformin exposure but are these effects any worse or better than what would be seen in women who had GDM but not treated with metformin?

While well written, the intro is a little long, the sections on metformin during pregnancy 79-110 could be condensed into one section.

Methods: What was the method of randomisation used?

Why was the metformin given prior to pregnancy 19days. In a human pregnancy complicated by GDM would the metformin not be applied later in pregnancy? I guess you need to be clear about whether you're modelling GDM or insulin resistance/T2DM during pregnancy. Maybe a statement to clarify would help.

Why was an overnight fast used rather than a 6 hour fast? Overnight is quite severe? Given that one sibling fasted and one not fasted, does this mean that they were separated the night before cull and would this induce stress in the animal?

It would be really useful to have the n-number of animals used in the animal model section of the methods with clear identification of the number of litters used for the offspring experiment. Rather than having it in the statistics section.

Statement of n-number would also be really useful in Figure 1.

In the statistic section n numbers are indeed mentioned, was a power calculation carried out? I'm just wondering if the reduction in numbers due to technical failures resulted in reduced power of the experiment?

The authors also mentioned that statistical outliers were removed, how were statistical outliers identified?

Results: Figure 2 line 301 (b) should be (f).

Can the authors just clarify for Table 1 than the same animals were used in these experiments at each of the 3 timepoints. The authors also mentioned that there can be issues with outliers and technical issues - was the n-number for the final data in table two the same for each group across the 3 timepoints.

Discussion:

Figure 5 is lovely, a really nice way to summarise the results.

Referee #2:

Metformin is the first-choice drug for use in treating diabetes and is increasingly being used to treat diabetes in pregnancy. There are clear benefits for mothers but the impact on the offspring is unclear. This work assessed the impact of maternal metformin treatment in mice, on cardiovascular health of 12-month-old offspring. It is important and timely.

Overall, the manuscript is well written, and the data are robust and support the study objectives and conclusions. The summary figure (Figure 5) is a good addition and is particularly helpful given the complexity of the data.

However, I have some minor comments:

1) In the introduction, or methodology, it would be useful to have information on what the equivalent age of 12-month-old mice would be for humans.

2) Line 352-authors state that 'Cardiac weight may be slightly increased in female Ob-Met offspring, but this did not reach statistical significance'. However, the * suggests statistical significance. Can the authors clarify this. Additionally, it would be interesting to assess whether this potential impact on heart weight was still present, or potentially more pronounced, when corrected for femur length as an indicator of cardiac hypertrophy. Similarly for male offspring (Figure 4I)

END OF COMMENTS

JP-RP-2025-288696 – Cardiovascular outcome in 12-month-old male and female offspring of metformin-treated obese mice

Comments from the editor, Laura Bennett

We thank the senior editor for the kind comments and were very pleased to hear the paper is accepted for publication upon acceptable revision. We have adhered to the general requirements, instructions for authors and revision checklist as outlined in this acceptance email (dated 2/4/2025). In addition, see our detailed responses to the comments below:

- Include a Key Points list in the article itself, before the Abstract. **This has now been added to the revised version of the manuscript.**
- Author photo and profile. First or joint first authors are asked to provide a short biography (no more than 100 words for one author or 150 words in total for joint first authors) and a portrait photograph. These should be uploaded and clearly labelled together in a Word document with the revised version of the manuscript. See Information for Authors for further details. **This has now been added to the revised version of the manuscript.**
- The reference list must be in alphabetical order, rather than numbered, to comply with our Journal format. **The references have been updated to the correct format.**
- The Journal of Physiology funds authors of provisionally accepted papers to use the premium BioRender site to create high resolution schematic figures. Follow this link and enter your details and the manuscript number to create and download figures. Upload these as the figure files for your revised submission. If you choose not to take up this offer, we require figures to be of similar quality and resolution. If you are opting out of this service to authors, state this in the Comments section on the Detailed Information page of the submission form. The link provided should only be used for the purposes of this submission. Authors will be charged for figures created on this premium BioRender account if they are not related to this manuscript submission. **We have recreated the Abstract Figure, Figure 1 and Figure 5 using BioRender Premium. The software has not been used to make figures unrelated to the manuscript.**
- Please upload separate high-quality figure files via the submission form. **We have exported high-quality figures and have uploaded these to the submission form.**
- Your paper contains Supporting Information of a type that we no longer publish, including supplementary tables and figures. Any information essential to an understanding of the paper must be included as part of the main manuscript and figures. The only Supporting Information that we publish are video and audio, 3D structures, program codes and large data files. Your revised paper will be returned to you if it does not adhere to our Supporting Information Guidelines. **Essential information, such as the qPCR primer sequences and echocardiography**

calculations used to generate the physiological data, has been moved from supplementary data to the main body of the text. We have added the large full datasets containing echocardiography, wire myography areas under or above the curves, and *ex vivo* molecular/histological analyses as Supporting Information, for full transparency and to aid in understanding of the main findings of the manuscript. All non-essential supplementary information as shown in the original manuscript has been removed.

- Papers must comply with the Statistics Policy: https://jp.msubmit.net/cgi-bin/main.plex?form_type=display_requirements#statistics. In summary:
 - a. If $n \leq 30$, all data points must be plotted in the figure in a way that reveals their range and distribution. A bar graph with data points overlaid, a box and whisker plot or a violin plot (preferably with data points included) are acceptable formats. **All bar and scatterplot graphs show individual datapoints. The one exception are the myography dose-response curves, where plotting the data for each animal at a given dose would hinder interpretation of the graph. This complies with the statistics policy as each animal subject has numerous data points associated with it, allowing 'n' to be treated as being each data point (in this case dose), not the number of subjects per dose per group.**
 - b. If $n > 30$, then the entire raw dataset must be made available either as supporting information, or hosted on a not-for-profit repository, e.g. FigShare, with access details provided in the manuscript. **The manuscript does not contain datasets with $n > 30$ per group, but for transparency the full datasets are included as supporting information.**
 - c. 'n' clearly defined (e.g. x cells from y slices in z animals) in the Methods. Authors should be mindful of pseudoreplication. **N-numbers for the whole study have been included in the methods. It is also highlighted in the methods that n relates to the number of litters represented.**
 - d. All relevant 'n' values must be clearly stated in the main text, figures and tables. **All figures, tables and legends contain information about n-numbers per group for each parameter.**
 - e. The most appropriate summary statistic (e.g. mean or median and standard deviation) must be used. Standard Error of the Mean (SEM) alone is not permitted. **We are committed to data transparency: we show each individual datapoint in our figures where possible, and have provided the full datasets including SD in the Supporting Information. However, although we agree that mean \pm SD (with *n*) is statistically the more appropriate, using mean \pm SEM makes it easier to compare our data to other published studies as it remains the most used method of reporting data in publications. Therefore, we request that we leave data**

within the main manuscript as mean \pm SEM, with information on the full data sets with SDs easily accessible in the supporting data files.

- f. Exact p values must be stated. Authors must not use 'greater than' or 'less than'. Exact p values must be stated to three significant figures even when 'no statistical significance' is claimed. **Exact p-values have been included in all figures and tables, except for p-values <0.0001 for which exact values could not be obtained (our analysis software only provides 4 decimal numbers for the p-values).**
- Please include an Abstract Figure file, as well as the Figure Legend text within the main article file. The Abstract Figure is a piece of artwork designed to give readers an immediate understanding of the research and should summarise the main conclusions. If possible, the image should be easily 'readable' from left to right or top to bottom. It should show the physiological relevance of the manuscript so readers can assess the importance and content of its findings. Abstract Figures should not merely recapitulate other figures in the manuscript. Please try to keep the diagram as simple as possible and without superfluous information that may distract from the main conclusion(s). Abstract Figures must be provided by authors no later than the revised manuscript stage and should be uploaded as a separate file during online submission labelled as File Type 'Abstract Figure'. Please also ensure that you include the figure legend in the main article file. All Abstract Figures should be created using BioRender. Authors should use The Journal's premium BioRender account to export high-resolution images. Details on how to use and access the premium account are included as part of this email. **We have added a summary figure to the manuscript. As we have received such lovely positive from the referees regarding Figure 5, we have used this as a template to generate a simplified version of this figure without recapitulating it exactly.**

Referee #1:

Schoonejans et al present a study which examines the impact of obese pregnancy plus or minus metformin exposure on 12 month old male and female offspring. This is a well written and nicely presented study. It is topical and will be of interest to the readers of this journal. I only really had a few minor queries.

We thank the referee for their kind comments and are glad they find our work of interest.

1. Intro: Well written intro. I just have one query. You describe human studies that you have outlined showing subtle cardiovascular effects in children who were exposed in utero to metformin. It is clear that they are impacted by the metformin exposure but are these effects any worse or better than what would be seen in women who had GDM but not treated with metformin? **We thank the referee for this comment and are happy to clarify. Unfortunately, no studies looking at cardiovascular outcomes in metformin-exposed children from pregnancies complicated by GDM have been performed to date, instead studies cited have been done in women with obesity but without GDM (Panagiotopoulou et al., 2020) and women with PCOS (Hanem et al., 2019). Panagiotopoulou et al. describe a shorter IVRT, smaller left atrial area and slight reduction in aortic (but not peripheral) pressure in metformin-exposed children. They note this could reflect improved diastolic relaxation and reduced cardiac/vascular stiffness, but as the values were in the normal range and these parameters are age-dependent they are cautious to make strong statements about whether this is beneficial or not. Hanem et al. describe a slightly clearer picture of potential worsening of cardiovascular parameters in metformin-exposed children of women with PCOS, specifically effects on systolic blood pressure and heart rate. With this in mind, we have opted not to describe these findings in detail, but have summarised them as follows: “Metformin administration in glucose tolerant obese pregnancies was associated with subtly altered haemodynamic and diastolic parameters in 4-year-old children (Panagiotopoulou et al., 2020), changes that if persistent may be beneficial for future cardiovascular health. In contrast, adverse outcomes were seen in 5- to 10-year-old children of metformin-treated mothers with PCOS, such as increased risk of “metabolically unhealthy” obesity (a composite outcome that included hypertension) and increased heart rate in male children (Hanem et al., 2019). However, children from these studies remain young, and further research is required to determine ageing-dependent effects in adult individuals prenatally exposed to metformin.”**
2. While well written, the intro is a little long, the sections on metformin during pregnancy 79-110 could be condensed into one section. **We thank the referee for this suggestion. The two sections have been merged and shortened by 100 words.**

3. Methods: What was the method of randomisation used? **The number of cages to be weaned onto specific diets (control or obesogenic) was determined by a researcher not involved in animal maintenance. This was communicated to a technician not involved in the research, who performed the randomisation. The same approach was taken when randomising for metformin treatment. We have made some clarifications in the text, e.g.: “Female C57Bl/6J mice (...) were randomly assigned, in numbers based on need, by a technician not involved in the research, to receive either a standard chow diet (...) or a high-fat, high-sugar obesogenic diet”.**
4. Why was the metformin given prior to pregnancy 19days. In a human pregnancy complicated by GDM would the metformin not be applied later in pregnancy? I guess you need to be clear about whether you're modelling GDM or insulin resistance/T2DM during pregnancy. Maybe a statement to clarify would help. **We are happy to provide further details about our model. We have previously shown that the impaired glucose tolerance in our model is pregnancy-associated and resolves after gestation, and thus more reflective of a gestational diabetic phenotype than a type 2 diabetic one (Fernandez-Twinn *et al.*, 2017; Hufnagel *et al.*, 2022; Schoonejans *et al.*, 2022). We also demonstrated that the impaired glucose tolerance in late gestation was improved by the metformin intervention and the increased fat mass and hepatic steatosis observed in the obese dams compared to the control-fed dams was reduced by metformin (Hufnagel *et al.*, 2022). This indicates that the mouse model recapitulates both impaired glucose tolerance during pregnancy and some of the major benefits of maternal metformin treatment seen in human trials. More information about the model has been added to the Animal model section. However, we acknowledge that the timing of the initiation of metformin treatment (one week prior to mating) differs from that typically used to treat gestational diabetes (usually diagnosed at 24-28 weeks gestation) and is more reflective what happens in pregnant women with type 2 diabetes or PCOS. We have acknowledged this in the text.**
5. Why was an overnight fast used rather than a 6 hour fast? Overnight is quite severe? **Overnight fasting was used to standardise metabolic parameters measured in the serum and to allow comparison with previous findings.**
6. Given that one sibling fasted and one not fasted, does this mean that they were separated the night before cull and would this induce stress in the animal? **We apologise if this was not clear. The referee is correct, the two adult siblings were separated overnight prior to tissue collection – we have now clarified this in the methods. In our experience, separating the siblings does not lead to significant stress. For example, we observe no differences in animal behaviour or in fed blood glucose levels between those who were separated from their cagemate and those who were not. In the 12-month-old cohort used for this manuscript.**

Figure A: blood glucose levels in 8-week-old mice from a cohort similar and contemporary to the mice used in this manuscript; P-value reflects results from a two-tailed student's t-test.

7. It would be really useful to have the n-number of animals used in the animal model section of the methods with clear identification of the number of litters used for the offspring experiment. Rather than having it in the statistics section. **We thank the referee for this suggestion and agree this information may be better suited in the animal model section. We have also added more information about the number of dams (and thus litters, as each dam provided only one litter) used.**
8. Statement of n-number would also be really useful in Figure 1. **We thank the reviewer for this suggestion and have added this information to Figure 1.**
9. In the statistic section n numbers are indeed mentioned, was a power calculation carried out? I'm just wondering if the reduction in numbers due to technical failures resulted in reduced power of the experiment? **We thank the referee for this comment. Power calculations were carried out using standard deviation and mean of previously obtained ejection fraction data at 8 weeks of age. Based on this analysis, a sample size of 10 per group would be able to detect a 20% difference in mean with significance of $p < 0.05$ at 90% power (GraphPad StatMate 2.00) at this age. Hence, for this study, $n = 12$ offspring per group were purposely bred to maximise power while keeping in line with the 3Rs Principle of Reduction regarding the use of animals. The exclusion of datapoints due to quality control and statistical outliers likely resulted in some loss of power. However, most parameters for which this was done had $n \geq 8/9$ per group. For female mitral valve measures at 3 months of age, numbers were further reduced ($n = 6-8$). Yet, this did not prevent significant changes from being observed in these parameters. As cardiac phenotypes tend to worsen with age, true group differences are expected to be more easily detected at 3, 6 and 12 months than at 8 weeks. We are thus confident that the study was sufficiently powered to detect meaningful differences, and that our approach has generated high quality data. However, we do not exclude the possibility that more subtle changes could have been detected if more mice were used.**
10. The authors also mentioned that statistical outliers were removed, how were statistical outliers identified? **Statistical outliers were determined via the Grubb's**

test performed in GraphPad Prism. This information has been added to the manuscript.

11. Results: Figure 2 line 301 (b) should be (f). **This has been amended.**
12. Can the authors just clarify for Table 1 than the same animals were used in these experiments at each of the 3 timepoints. **We apologise that this was not clear from the manuscript in its additional form. Blood pressure and heart rate measurements were performed in the same animals at each time. This has been clarified in the title and legend of Table 1 (now Table 3 in the new version of the manuscript).**
13. The authors also mentioned that there can be issues with outliers and technical issues - was the n-number for the final data in table two the same for each group across the 3 timepoints. **Data from some animals had to be excluded due to quality control measures, leading to a slight difference in n-numbers between ages studied, e.g. when fewer than 10 measurements could be taken for each animal or when within-animal variation exceeded 15% for the SBP/HR measurement at that age. All data that passed quality control was then subjected to outlier analysis according to methods added to the statistical analysis section. We have added details about the quality control for the tail cuff plethysmography to the methods.**
14. Figure 5 is lovely, a really nice way to summarise the results. **Thank you very much, it is really appreciated.**

References in response to Referee 1

- Fernandez-Twinn DS, Gascoin G, Musial B, Carr S, Duque-Guimaraes D, Blackmore HL, Alfaradhi MZ, Loche E, Sferruzzi-Perri AN, Fowden AL & Ozanne SE (2017). Exercise rescues obese mothers' insulin sensitivity, placental hypoxia and male offspring insulin sensitivity. *Sci Rep* **7**, 44650.
- Hanem LGE, Salvesen Ø, Juliusson PB, Carlsen SM, Nossum MCF, Vaage MØ, Ødegård R & Vanky E (2019). Intrauterine metformin exposure and offspring cardiometabolic risk factors (PedMet study): a 5–10 year follow-up of the PregMet randomised controlled trial. *Lancet Child Adolesc Health* **3**, 166–174.
- Hufnagel A, Fernandez-Twinn DS, Blackmore HL, Ashmore TJ, Heaton RA, Jenkins B, Koulman A, Hargreaves IP, Aiken CE & Ozanne SE (2022). Maternal but not fetoplacental health can be improved by metformin in a murine diet-induced model of maternal obesity and glucose intolerance. *Journal of Physiology* **600**, 903–919.
- Panagiotopoulou O, Syngelaki A, Georgiopoulos G, Simpson J, Akolekar R, Shehata H, Nicolaidis K & Charakida M (2020). Metformin use in obese mothers is associated with improved cardiovascular profile in the offspring. *Am J Obstet Gynecol* **223**, 246.e1-246.e10.

Schoonejans JM, Blackmore HL, Ashmore TJ, Pantaleão LC, Pellegrini Pisani L, Dearden L, Tadross JA, Aiken CE, Fernandez-Twinn DS & Ozanne SE (2022). Sex-specific effects of maternal metformin intervention during glucose-intolerant obese pregnancy on body composition and metabolic health in aged mouse offspring. *Diabetologia* **65**, 2132–2145.

Referee #2:

Metformin is the first-choice drug for use in treating diabetes and is increasingly being used to treat diabetes in pregnancy. There are clear benefits for mothers but the impact on the offspring is unclear. This work assessed the impact of maternal metformin treatment in mice, on cardiovascular health of 12-month-old offspring. It is important and timely.

Overall, the manuscript is well written, and the data are robust and support the study objectives and conclusions. The summary figure (Figure 5) is a good addition and is particularly helpful given the complexity of the data.

We thank the referee for their kind comments and are glad they find our work robust and of interest.

1. In the introduction, or methodology, it would be useful to have information on what the equivalent age of 12-month-old mice would be for humans. **We thank the referee for this helpful suggestion. In mice, 12 months equates to middle age (normal life span is 2 years). This information has been added to the introduction.**
2. Line 352-authors state that 'Cardiac weight may be slightly increased in female Ob-Met offspring, but this did not reach statistical significance'. However, the * suggests statistical significance. Can the authors clarify this. **Statistical analysis showed that there was a trend for cardiac weight to be increased, by which we mean $0.05 > p > 0.1$. This does not meet our statistical threshold but suggests a potential effect that would need to be confirmed in repeat experiments. In the revised version of the manuscript the (*) is replaced by the actual p-value.**
3. Additionally, it would be interesting to assess whether this potential impact on heart weight was still present, or potentially more pronounced, when corrected for femur length as an indicator of cardiac hypertrophy. Similarly for male offspring (Figure 4I). **We agree with the referee that this would give important information about the nature of the hypertrophy, and in future studies this will be done. Unfortunately, femur length was not collected in this study.**

Dear Dr Schoonejans,

Re: JP-RP-2025-288696R1 "**Cardiovascular outcome in 12-month-old male and female offspring of metformin-treated obese mice**" by Josca Mariëtte Schoonejans, Phoebe Wilshire, Lais Vales Mennitti, Kwun Kiu Wong, Thomas J Ashmore, Tessa A.C. Garrud, Heather Louise Blackmore, Olga Patey, Denise S Fernandez-Twinn, Dino A Giussani, and Susan E Ozanne

We are pleased to tell you that your paper has been accepted for publication in The Journal of Physiology.

Yours sincerely,

Laura Bennet
Senior Editor
The Journal of Physiology

If you would like to receive our 'Research Roundup', a monthly newsletter highlighting the cutting-edge research published in The Physiological Society's family of journals (The Journal of Physiology, Experimental Physiology, Physiological Reports, The Journal of Nutritional Physiology and The Journal of Precision Medicine: Health and Disease), please click this link, fill in your name and email address and select 'Research Roundup':
<https://www.physoc.org/journals-and-media/membernews>

- You can help your research get the attention it deserves! Check out Wiley's free Promotion Guide for best-practice recommendations for promoting your work at: www.wileyauthors.com/eeo/guide. You can learn more about Wiley Editing Services which offers professional video, design, and writing services to create shareable video abstracts, infographics, conference posters, lay summaries, and research news stories for your research at: www.wileyauthors.com/eeo/promotion.

EDITOR COMMENTS

Reviewing Editor:

Thank you for the revisions. Congratulations on an important contribution to the field and the Journal.

REFeree COMMENTS

Referee #1:

I am happy that the authors have appropriately addressed by comments. Well done on a lovely paper.

Referee #2:

The authors have provided appropriate and thoughtful response to my comments. I am happy that all comments have been addressed appropriately and I have no further suggestions.